# Global Motif Embedding Meet Few-Shot Graph Learning

## Abstract

Graph Neural Networks (GNNs) have shown strong performance in node classification tasks. However, in real-world scenarios, only a limited number of nodes are often labeled, leading to the few-shot node classification problem, which is a significant challenge for GNNs. Most existing research focuses on designing new models to adapt to this setting, but often overlooks structural information in the graph data, such as motif patterns, which can provide crucial cues for learning from a few examples. In this paper, we propose a novel framework that integrates motif representations into graph few-shot learning models. Specifically, we extract unique motif representations from the graph and introduce them as virtual nodes. To capture richer structural patterns, we further enhance motif extraction by adding cluster labels based on node similarity, thereby incorporating both structural and feature information. Additionally, we assign TF-IDF scores as edge weights between virtual motif nodes and original nodes to quantify the importance of their connections. Experimental results demonstrate that our approach consistently improves the performance of various graph few-shot learning methods.

## 1 Introduction

Recently, Graph Neural Networks (GNNs) have achieved remarkable success in graph-related tasks such as node classification and link prediction. However, in real-world scenarios, graphs are often sparsely labeled or lack sufficient annotations, which poses a significant challenge for GNN-based models. This problem setting is commonly referred to as graph few-shot learning. To address this, several graph few-shot learning models have been proposed, demonstrating promising performance in improving GNNs under limited supervision.

Most existing research has primarily focused on improving the model architecture itself to achieve better performance. A common direction is the application of meta-learning frameworks (Kim et al., 2023; Huang & Zitnik, 2021; Wang et al., 2022; 2023a). Some studies further extend this line of work by integrating contrastive learning with meta-learning frameworks (Wang et al., 2023b; Liu et al., 2025a; 2024). However, these approaches largely overlook an important aspect: even when label information is limited, the graph structure itself still contains latent relational information, such as node connections and degree distributions. Therefore, it is crucial to exploit such information from the original graph, as it can provide additional support for training graph few-shot learning models.

In NLP tasks, researchers often encode frequently occurring words or subwords as unique embeddings, known as word embeddings (Mikolov et al., 2013). These embeddings can then be leveraged by language model to better understand input sentences. In the field of graph learning, recurring connection patterns in a graph are referred to as motifs (Milo et al., 2002). Inspired by word embeddings, we introduce motif embeddings as an additional component for graph few-shot learning. However, several challenges remain. First, there is currently no framework to effectively integrate motif embeddings with the original graph. Second, when introducing motifs into FSL, it is difficult to control the number of motif patterns or potential extra structure that are incorporated.

To address these limitations, we propose a novel framework called MoEFL (Motif-Embedding-Few-shot-Learning), which incorporates motif information into graph few-shot learning through learnable motif embeddings in a lightweight and modular design. Instead of modifying the backbone encoder directly, MoEFL

introduces a separate motif encoding branch that augments node representations with motif-aware structural features.

Specifically, each extracted motif is represented as a learnable embedding, serving as a globally shared structural representation across the graph. These embeddings capture recurring structural patterns and enable knowledge transfer across different regions of the graph, analogous to word embeddings in language models. To enhance their expressiveness, we introduce cluster labeling, which assigns temporary labels to nodes based on feature similarity, allowing the model to distinguish structurally similar but semantically different motifs without increasing motif size. In addition, we adopt a TF-IDF-based weighting scheme to quantify the importance of motif–node connections, effectively regulating the number and influence of virtual edges introduced into the graph.

In summary, our contributions are as follows:

- We propose a motif embedding framework that represents motifs as learnable embeddings, enabling them to be co-trained during few-shot learning, integrated into the original graph as virtual nodes, and transferred to new tasks.

- We introduce cluster labeling and TF-IDF weighting to control motif diversity and regulate the number of virtual edges.

- We demonstrate that our framework consistently improves performance over two state-of-the-art graph few-shot learning methods.

## 2 Related Work

### 2.1 Graph few-shot learning

Graph few-shot learning aims to enhance the performance of graph neural networks (GNNs) (Xu et al., 2019) when labeled data is scarce. A common strategy is to integrate GNNs with meta-learning frameworks such as MAML (Finn et al., 2017; Rajeswaran et al., 2019) and ProtoNetSnell et al. (2017). Several studies have explored this integration from different perspectives. For example, G-META (Huang & Zitnik, 2021) trains models on local subgraphs surrounding the target node to extract more relevant information. Tent (Wang et al., 2022) reduces variance across nodes, classes, and tasks, while TEG (Kim et al., 2023) introduces task-equivariant graphs that adapt to transformations and capture transferable patterns. (Liu et al., 2025c) introduces spectrums experts to adapt to local graph structure under different FSL tasks. In parallel, contrastive learning has also been employed to boost performance. COSMIC (Wang et al., 2023b) is the first to incorporate contrastive learning for better generalization on novel classes, STAR (Liu et al., 2025a) further refines this approach by introducing optimal transport and set representation. In addition, (Liu et al., 2025b) focuses on improving model performance under limited training tasks.

### 2.2 Motif in graph

A motif is defined as a recurring subgraph pattern that frequently appears in a network (Milo et al., 2002). Motifs have been extensively studied in graph-related research, particularly in the chemistry domain (Das & Dai, 2007; Abou Assi et al., 2018; Thandapani et al., 2013; Kumar et al., 2022). Recently, motifs have also been incorporated into graph learning models. For example, (Yu & Gao, 2025) leverage motifs to improve the interpretability of GNN outputs, while (Zhong et al., 2024) integrate motifs into transformer architectures for drug–drug interaction prediction. However, these approaches primarily focus on chemistry-related graphs and are difficult to generalize to broader domains such as social networks or citation graphs.

Several works have explored motif integration in general graphs. For instance, (Chen et al., 2023) construct motif-based adjacency matrices using 3-node structural motifs to enhance GNN performance. Nevertheless, such methods rely solely on graph topology and simple node-level structures, resulting in limited motif expressiveness. To address this limitation, we propose a motif-aware framework for graph few-shot learning that incorporates cluster labeling to enrich motif representations.

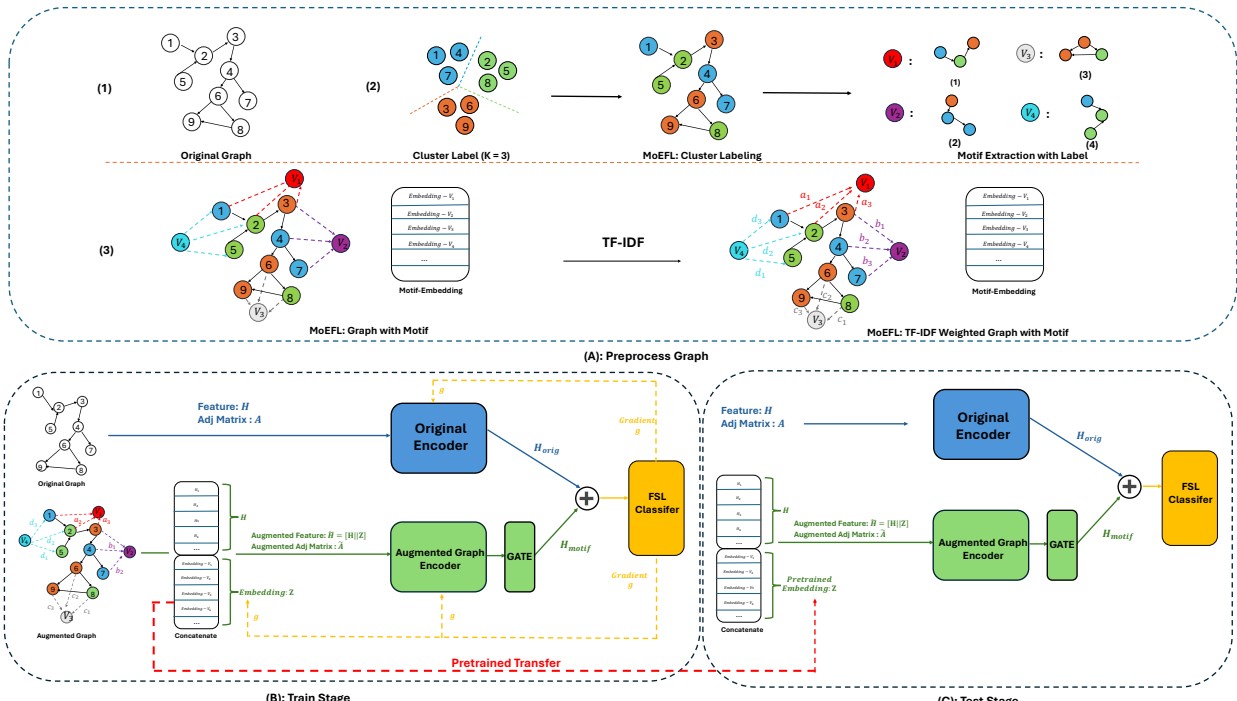

Figure 1: Overview of the proposed framework. (A) In the preprocessing stage, we extract cluster-labeled motifs and associate each motif with a learnable embedding. (B) In the training stage, the motif embeddings are jointly optimized with both the original graph encoder and the augmented graph encoder. (C) In the test stage, the learned motif embeddings are reused as pretrained representations for new tasks without further optimization.

## 2.3 Word Embedding

Word embeddings (Mikolov et al., 2013) are widely used in natural language processing as low-dimensional vector representations of words. In such embedding spaces, words with similar semantic meanings are mapped to nearby vectors. GloVe (Pennington et al., 2014) further extends this idea by incorporating global word co-occurrence statistics into the embedding learning process. Inspired by these works, we introduce globally shared motif embeddings into graph few-shot learning as an auxiliary source of global structural features.

## 3 Motif for Graph Few-shot learning

In this section, we introduce our proposed framework **MoEFL**, which is **Mo**tif-**E**mbedding-**F**ew-shot-**L**earning framework.

### 3.1 Preliminary

Given a graph $G$ that is represented by $\{V, \epsilon\}$, where the $V$ denote the vertex $\{V_1......V_n\}$, and $\epsilon$ denotes the edge $\{\epsilon_1......\epsilon_n\}$. In this paper, we focus on the few-shot node classification problem, where only a limited number of node labels are available during the training stage. To enable effective model learning under this constraint, a general way is to adopt a meta-learning framework that constructs a large number of tasks. Each task is defined by $N$ classes, with each class containing $K$ nodes, following the standard "N-way-K-shot" formulation. During the test phase, evaluation is conducted on tasks with the same " N-way-K-shot" setting but involving novel classes that were not observed during training.

## 3.2 Motivation and Challenge

Graph few-shot learning is fundamentally challenged by the scarcity of labeled data, which limits the ability of graph neural networks (GNNs) to learn discriminative representations. In natural language processing (NLP), pretrained word embeddings have proven highly effective in addressing this issue by learning representations from large-scale unlabeled corpora and transferring them to downstream tasks with limited supervision. Such embeddings capture semantic similarity between words, enabling models to generalize by mapping words with similar meanings to nearby representations in the embedding space.

In this work, we adopt the motif definition and extraction procedures from prior studies such as Chen et al. (2023); Zhao et al. (2025). Specifically, motifs are defined as frequently occurring subgraph patterns, which serve as fundamental structural primitives in graphs. While we do not modify the underlying extraction process, we build upon these motifs to explore their role in graph few-shot learning. By capturing reusable local structures, motifs provide informative inductive biases for representation learning, particularly in low-data regimes. However, despite their potential, the use of motifs in graph few-shot learning remains largely underexplored.

Several challenges hinder the effective use of motifs in this setting. First, there is no well-established pre-training paradigm for learning motif-level embeddings that can be transferred to downstream tasks. Second, conventional motif extraction methods rely solely on graph topology and ignore node features, treating structurally identical subgraphs as equivalent even when they carry different semantic meanings. This significantly limits the expressiveness of motif-based representations, especially in few-shot scenarios where feature-dependent patterns are critical for generalization.

These limitations highlight the need for a unified framework that can effectively incorporate motif-level information into graph few-shot learning. To this end, we propose MoEFL, a principled framework that models motifs as learnable embeddings and integrates both structural and feature-aware information to enhance representation learning under limited supervision.

## 3.3 Motif embedding to enhance graph few-shot learning

We now introduce the MoEFL framework, which addresses the first challenge, the lack of pretrained motif representations, by modeling motifs as learnable embeddings that can be integrated into the learning process.

We follow the basic definition for motifs which are adopted by previous research works including Chen et al. (2023); Zhao et al. (2025). The motif is defined as a frequently occurring subgraph pattern centered around a node and its local neighborhood. While motifs capture important structural regularities, they lack inherent vector representations, which largely limit the application of motif in downstream learning.

Some existing approaches attempt to derive simple motif representations using frequency-based methods. For example, (Yu & Gao, 2022; Budowski-Tal et al., 2010) adopt bag-of-words style representations, where motifs are treated as discrete tokens and embedded based on their occurrence statistics. However, such methods fail to explore the semantic meaning between motifs, while just treat them as single words, which the motif is treated as discrete tokens based on their frequency.

To address this limitation, we introduce pretrained motif embeddings that enable motifs to directly participate in representation learning and facilitate knowledge transfer in few-shot settings. By mapping motifs into a continuous embedding space, our approach captures structural similarity and relational patterns between motifs, thereby overcoming the limitations of bag-of-words representations. Specifically, the motif embeddings $Z$ are learned during the training stage on the source graph, where they are jointly optimized with downstream objectives to encode transferable structural knowledge. These embeddings, learned during the training stage, are then reused as pretrained representations without further updating for new tasks with limited supervision, providing a strong initialization and improving generalization performance.

Specifically, inspired by word embedding methods Mikolov et al. (2013); Pennington et al. (2014), each motif is associated with a dense embedding vector $Z$, which serves as a globally shared structural representation across the graph. In this formulation, motifs function analogously to words in NLP: each motif is treated as a token and mapped to a continuous vector representation in the embedding space. This enables the model

to capture similarities and relationships between motifs, allowing it to generalize across structurally related subgraphs while preserving distinctions among different motif patterns.

### 3.3.1 Motif Interaction

To incorporate motif embeddings into the graph, we introduce a set of virtual nodes $V'$, where each virtual node corresponds to a unique motif and is parameterized by its embedding in $Z$. An original node is connected to a virtual node if it participates in a subgraph instance matching the corresponding motif. This construction provides a unified and model-agnostic mechanism for integrating global structural information into the graph. As a result, we obtain an augmented adjacency matrix $\tilde{A}$, which are defined as follows:

$$\tilde{A} = \begin{bmatrix} A & B \\ B^\top & 0 \end{bmatrix} \in \mathbb{R}^{(N+M)\times(N+M)}. \tag{1}$$

To model the interaction between node features and motif information, we adopt a dual-branch architecture. Given the original node features $X \in \mathbb{R}^{N \times d_0}$ and adjacency matrix $A \in \mathbb{R}^{N \times N}$, we first obtain node representations using the backbone encoder on the original graph:

$$H_{\text{orig}} = f_{\text{enc}}(X, A), \quad H_{\text{orig}} \in \mathbb{R}^{N \times d}, \tag{2}$$

where $f_{\text{enc}}$ denotes the encoder used in the downstream model.

In parallel, the augmented graph is encoded through a separate graph encoder operating on augmented graph $\tilde{A}$, which captures the relationships between nodes and motif instances. Specifically, we first project the original node features into a higher-dimensional space, and concatenate them with the learnable motif embeddings $Z$ to form the input of the motif-aware graph, where the final augmented feature embedding defined as

$$H' = \text{MLP}(X), \quad H' \in \mathbb{R}^{N \times d}, \tag{3}$$

$$\tilde{X} = [H' \parallel Z] \in \mathbb{R}^{(N+M) \times d} \tag{4}$$

A graph neural network is then applied to model interactions between nodes and motifs:

$$\tilde{H} = \text{GNN}\left([H' \parallel Z], \tilde{A}\right), \quad \tilde{H} \in \mathbb{R}^{(N+M) \times d}. \tag{5}$$

We retain only the first $N$ rows corresponding to the original nodes, yielding the motif-enhanced representations as $H_{\text{aug}} = \tilde{H}_{1:N} \in \mathbb{R}^{N \times d}$.

To integrate the two representations, we adopt a residual fusion mechanism:

$$H = H_{\text{orig}} + \gamma \cdot H_{\text{aug}}, \quad H \in \mathbb{R}^{N \times d}, \tag{6}$$

where $\gamma \in R^d$ is a learnable gate vector that controls the contribution of motif information.The figure 1 depicts the framework process.

### 3.3.2 Train Motif Embedding

During training, the backbone encoder, augmented graph encoder, and motif embeddings are optimized jointly in an end-to-end manner. We apply a downstream few-shot learning objective on the final representations $H$. Specifically, a classifier is trained on top of $H$, and the loss is computed based on the performance of the few-shot tasks. The gradients from this objective are backpropagated through both the backbone encoder and the motif encoder, allowing the motif embeddings $Z$ to be learned simultaneously with the model parameters. Once trained, the motif embeddings capture transferable structural patterns and can be directly reused during the few-shot testing phase without further adaptation, serving as a global pretrained structural prior for unseen tasks.

This joint optimization enables the motif embeddings to capture task-relevant structural patterns, while the residual fusion allows the model to adaptively balance feature-based and motif-based information during training.

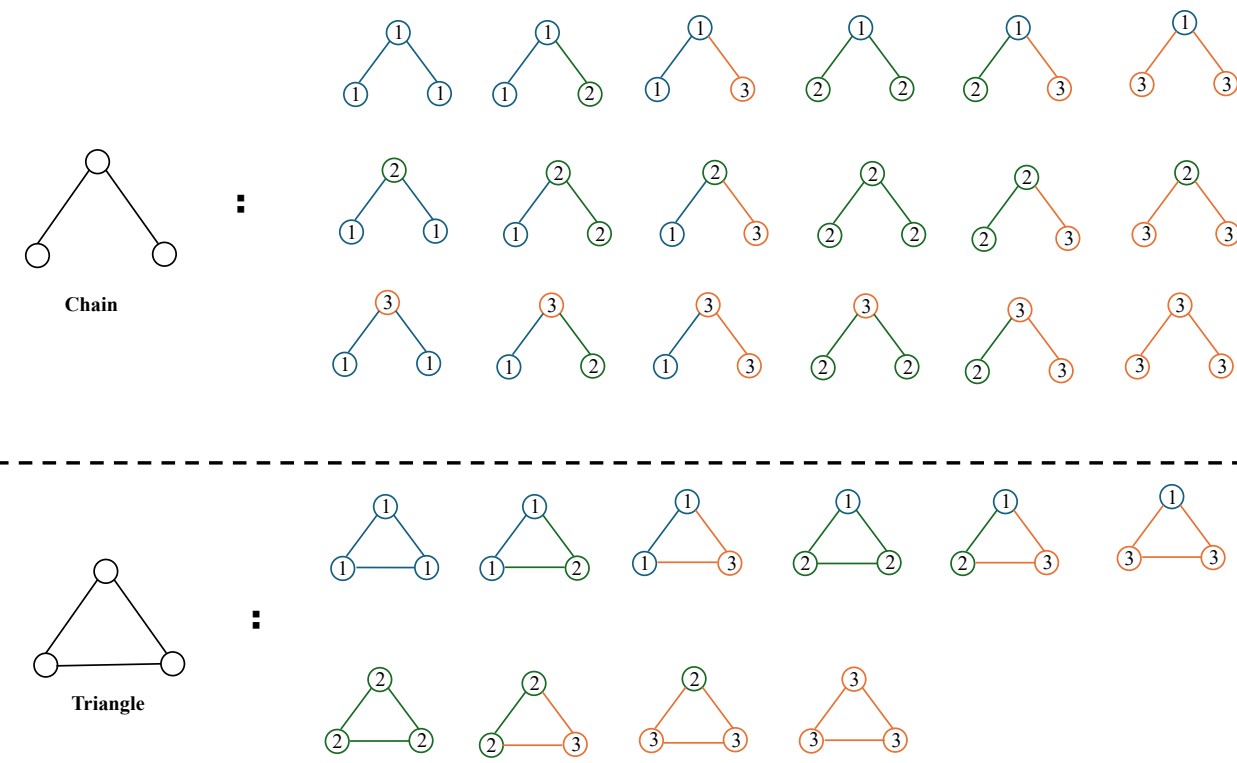

Figure 2: Illustration of motifs with node labels. In an undirected graph without node labels, only two types of 3-node motifs exist: a chain and a triangle. With a cluster label 3, the number of distinguishable motifs increases to 18 for chain-shaped motifs and 10 for triangle-shaped motifs.

### 3.4 Expand Motif via cluster labeling

In the previous section, we introduced the MoEFL framework, which incorporates globally shared motif embeddings as auxiliary information for graph few-shot learning and addresses the first challenge of missing motif representations. However, the effectiveness of this framework largely depends on the quality and diversity of the extracted motifs. In particular, the second challenge—limited expressiveness of conventional motif construction—is reflected in the following limitations of existing approaches. Methods such as (Chen et al., 2023) rely solely on structural patterns, which introduces several issues when applied in our setting.

- First, restricting motifs to purely structural patterns significantly limits their expressiveness. For example, prior work (Chen et al., 2023) shows that there are only a small number of possible motif patterns (e.g., 13 in directed graphs), which reduces further (to 2) in undirected settings commonly used in GNNs. Moreover, the number of motif types is fixed and does not scale with graph size, constraining the diversity of structural information that can be captured.

- Second, conventional motif extraction relies solely on topology and ignores node features, which are crucial for graph learning. Nodes with identical structural roles may carry very different semantic meanings due to their features, yet structure-only motifs treat them as equivalent.

These limitations significantly reduce motif diversity and hinder the effectiveness of the proposed framework.

To address this limitation, we propose incorporating node feature information into the motif extraction process. In the few-shot setting, where class labels are scarce, node feature similarity provides a key signal for capturing relationships between nodes. To leverage this signal, we cluster all nodes in the original graph into $K$ clusters based on feature similarity and assign each node a temporary label according to its cluster membership. These cluster-based labels serve as surrogate attributes, enabling the extraction of more diverse

and informative motifs beyond purely structural patterns. In this paper, we use Kmeans as our clustering algorithm. However, MoEFL is not tied to a specific clustering algorithm; alternative methods can be used without changing the framework.

As illustrated in Figure 2, the introduction of cluster-based labels significantly increases motif diversity. For example, in the case of undirected 3-node motifs, clustering the nodes of the original graph into three groups (without considering edge directions) expands the number of possible motif patterns to a theoretical total of 28. This expansion substantially increases the representational capacity of motifs, allowing structurally similar subgraphs to be distinguished based on feature context.

These motifs capture richer interactions beyond simple connectivity. For instance, the motif $1\rightarrow2\rightarrow3$ may represent a fundamentally different interaction pattern from $1\rightarrow3\rightarrow2$, even though they share the same underlying topology. Unlike traditional motif extraction methods, our approach distinguishes such patterns effectively. Importantly, this method does not rely on ground-truth labels and provides flexible control over the granularity of extracted patterns through the choice of the number of clusters. In Section 4.2.2, we further validate its effectiveness through comparison with conventional motif extraction methods.

### 3.5 Weighted Edge between Motif and Original Node

Within the MoEFL framework, introducing motif-based virtual nodes increases graph connectivity, as each virtual node may connect to many original nodes. In extreme cases, a single motif can introduce a large number of edges, potentially dominating the message-passing process and weakening the contribution of original node features.

To mitigate this issue, we assign importance weights to node–motif edges based on their relevance. Specifically, we adopt a TF–IDF-inspired weighting scheme tailored to the graph setting.

The term frequency (TF) measures how frequently a node participates in instances of a given motif:

$$TF(v,m) = \log\big(\#\mathrm{Iso}(\mathrm{subgraph}(v), m)\big), \tag{7}$$

where $\#\mathrm{Iso}(\mathrm{subgraph}(v), m)$ denotes the number of occurrences of motif $m$ centered at node $v$.

The inverse document frequency (IDF) reflects how broadly a motif appears across the graph:

$$IDF(m) = \frac{1}{|\{v \mid v \text{ participates in } m\}|}. \tag{8}$$

The final TF–IDF score is defined as:

$$w_{v,m} = TF(v,m) \cdot IDF(m), \tag{9}$$

which is used as the weight for the edge between node $v$ and motif $m$.

This weighting scheme downweights overly common motifs while preserving informative ones. In addition, motifs with very low relevance are filtered out, reducing unnecessary connections and keeping the computational overhead manageable.

## 4 Experiment

In this section, we conduct various experiments to prove the effectiveness of our framework.

### 4.1 Dataset Introduction

In this work, we follow the setting of Liu et al. (2025c) and choose to work on following datasets Cora (Yang et al., 2016), Citeseer (Yang et al., 2016), Amazon-Computer (Shchur et al., 2018), CoauthorCS (Bojchevski & Günnemann, 2018),CoraFull (Bojchevski & Günnemann, 2018), DBLP (Tang et al., 2008) and ogbn-arxiv (Hu et al., 2021). These include four small datasets and two large datasets. The detail of dataset is included in Table 1.

Table 1: Statistics of the evaluated datasets.

| Dataset | #Nodes | #Edges |
|---|---|---|
| Cora | 2,708 | 5,278 |
| CiteSeer | 3,327 | 4,552 |
| Amazon-Computer | 13,381 | 245,778 |
| Coauthor-CS | 18,333 | 81,894 |
| CoraFull | 19,793 | 65,311 |
| DBLP | 40,672 | 144,135 |
| ogbn-arxiv | 169,343 | 1,166,243 |

Table 2: Accuracies (%) on the Cora, CiteSeer and Amazon-Computer. The NwKs represents N-way-K-shot.

| Model | Cora | | | CiteSeer | | | Amazon-Computer | | |
|---|---|---|---|---|---|---|---|---|---|
| | 2w1s | 2w3s | 2w5s | 2w1s | 2w3s | 2w5s | 2w1s | 2w3s | 2w5s |
| G-Meta | 59.72±3.15 | 74.39±2.69 | 80.05±1.98 | 54.39±2.19 | 57.59±2.42 | 62.49±2.30 | 64.56±3.10 | 69.49±2.42 | 73.50±2.92 |
| TENT | 55.39±2.16 | 58.25±2.23 | 66.75±2.19 | 60.03±3.11 | 65.20±3.19 | 67.59±2.95 | 80.75±2.95 | 85.32±2.10 | 89.22±2.16 |
| Meta-GPS | 62.19±2.12 | 80.29±2.15 | 83.79±2.10 | 58.95±2.12 | 69.95±2.02 | 72.56±2.06 | 82.12±2.55 | 87.10±2.65 | 90.16±2.05 |
| X-FNC | 61.47±2.99 | 78.19±3.25 | 82.70±3.19 | 58.79±2.56 | 67.96±3.10 | 70.29±3.05 | 81.50±2.29 | 86.39±2.29 | 90.25±2.26 |
| TEG | 62.52±2.95 | 80.65±1.53 | 84.50±2.01 | 59.70±2.69 | 73.79±1.59 | 76.79±2.12 | 86.49±2.10 | 89.02±2.57 | 92.40±2.05 |
| COSMIC | 63.16±2.47 | 65.37±2.49 | 69.10±2.30 | 60.95±2.75 | 70.22±2.56 | 75.10±2.30 | 85.49±2.46 | 88.26±2.02 | 91.59±2.59 |
| STAR | 84.83±4.50 | 86.37±2.13 | 88.87±1.68 | 69.63±1.95 | 76.50±2.12 | 79.43±0.45 | 84.60±2.99 | 93.80±3.85 | 93.47±1.25 |
| **STAR+MoEFL** | **85.33±3.49** | **87.93±2.28** | **91.77±1.73** | **76.73±2.35** | **78.13±1.30** | **80.10±1.22** | **90.33±2.85** | **94.33±2.80** | **96.40±0.43** |
| GRACE | 66.48±2.88 | 82.40±2.03 | 86.19±1.80 | 63.90±2.84 | 75.67±2.44 | 79.64±1.79 | 90.23±0.90 | 92.46±0.55 | 94.66±0.50 |
| **GRACE+MoEFL** | **68.74±2.90** | **83.08±2.10** | **87.22±1.70** | **69.17±2.97** | **79.24±2.04** | **80.61±1.89** | **95.38±0.71** | **96.68±1.68** | **97.22±0.99** |

For graph few-shot learning, there are two main methodological paradigms: **regular meta-learning** and **meta-learning with contrastive learning**. In this work, we evaluate our method on two most recent state-of-the-art backbones, Grace (Liu et al., 2025c) and STAR (Liu et al., 2025a). For completeness, we also report the performance of several representative baselines, including G-Meta (Huang & Zitnik, 2021), TENT (Wang et al., 2022), X-FNC (Wang et al., 2023a), TLP (Tan et al., 2022), COSMIC (Wang et al., 2023b), and Meta-GPS (Liu et al., 2022) . The abbreviation **NwKs** represents $N$-way-$K$-shot.

Tables 2, 3 and 4 summarize the performance of our method across both small-scale and large-scale graph datasets.

On small datasets (Cora, CiteSeer, and Amazon-Computer), we observe that incorporating MoEFL consistently improves the performance of both STAR and GRACE backbones across different few-shot settings. In particular, under the 2-way settings, MoEFL yields consistent gains in both low-shot (1-shot) and higher-shot (3-shot, 5-shot) regimes. For example, compared to the vanilla GRACE backbone, MoEFL achieves noticeable improvements across all three datasets, with gains being more pronounced in lower-shot scenarios, indicating that motif-level structural information is especially beneficial when labeled data is scarce.

Similarly, when applied to the STAR backbone, our framework also leads to consistent performance improvements across all datasets and settings. Notably, the improvements are stable across different shot numbers, demonstrating that the benefit of MoEFL is not limited to a specific regime but generalizes across varying task difficulties.

On larger datasets, including Coauthor-CS, DBLP, CoraFull, and ogbn-arxiv, we observe a similar trend. MoEFL consistently enhances the performance of both STAR and GRACE across all evaluated settings. Although the magnitude of improvement varies depending on the dataset and task configuration, the gains remain consistent across different shot numbers and dataset scales, demonstrating the robustness of the proposed framework.

Overall, these results indicate that MoEFL provides a general and effective enhancement to graph few-shot learning models. By introducing motif-level structural representations, our method captures complementary information beyond standard node features and local message passing. The use of cluster-based motif construction further improves expressiveness, while the weighting mechanism helps regulate the contribution of motif connections. Together, these components enable MoEFL to integrate seamlessly with different backbone architectures and consistently improve their performance across diverse datasets.

Table 3: Accuracies (%) on the Coauthor-CS and DBLP. The NwKs represents N-way-K-shot

| Model | Coauthor-CS | | | | DBLP | | | |
|---|---|---|---|---|---|---|---|---|
| | 2w3s | 2w5s | 5w3s | 5w5s | 5w3s | 5w5s | 10w3s | 10w5s |
| G-Meta | 92.14±3.90 | 93.90±3.18 | 75.72±3.59 | 74.18±3.29 | 76.49±3.29 | 80.12±2.46 | 68.95±2.70 | 72.19±2.11 |
| TENT | 89.35±4.49 | 90.90±4.24 | 78.38±5.21 | 78.56±4.42 | 78.22±2.10 | 81.30±2.02 | 69.52±2.16 | 73.20±1.95 |
| Meta-GPS | 90.16±2.72 | 92.39±1.66 | 81.39±2.35 | 83.66±1.79 | 79.12±1.92 | 81.66±2.16 | 70.16±2.20 | 73.59±1.86 |
| X-FNC | 90.95±4.29 | 92.03±4.14 | 82.93±2.02 | 84.36±3.49 | 77.45±2.39 | 80.69±2.52 | 69.72±2.39 | 72.95±1.76 |
| TEG | 92.36±1.59 | 93.02±1.24 | 80.78±1.40 | 84.70±1.42 | 79.26±2.49 | 82.19±2.40 | 72.49±2.12 | 73.99±2.55 |
| COSMIC | 89.35±4.49 | 93.32±1.93 | 78.38±5.21 | 85.47±1.42 | 78.34±2.06 | 81.81±2.05 | 66.53±1.54 | 70.09±1.53 |
| **STAR** | 95.20±1.36 | **96.43±0.47** | 88.08±0.26 | 90.59±0.35 | 79.44±0.80 | 82.14±0.32 | 67.99±0.19 | 70.70±0.56 |
| **STAR+MOEFL** | **96.40±1.91** | 96.37±1.97 | **93.28±0.71** | **94.43±0.36** | **80.15±0.40** | **83.08±0.48** | **68.73±0.19** | **71.31±0.81** |
| **GRACE** | 95.50±1.30 | **96.20±0.97** | 86.03±1.05 | 86.82±1.01 | 81.72±2.05 | 85.30±1.90 | 74.22±1.56 | 76.70±1.46 |
| **GRACE+MOEFL** | 95.58±1.02 | 96.19±0.93 | **91.13±1.80** | **89.66±0.84** | **85.32±2.04** | **86.34±1.74** | **77.32±1.59** | **79.62±1.51** |

Table 4: Accuracies (%) on CoraFull and ogbn-arxiv. The NwKs represents N-way-K-shot.

| Model | CoraFull | | | | ogbn-arxiv | | | |
|---|---|---|---|---|---|---|---|---|
| | 5w3s | 5w5s | 10w3s | 10w5s | 5w3s | 5w5s | 10w3s | 10w5s |
| G-Meta | 57.52±3.91 | 62.43±3.11 | 53.92±2.91 | 58.10±3.02 | 40.48±1.70 | 47.16±1.73 | 35.49±2.12 | 40.95±2.70 |
| TENT | 64.80±4.10 | 69.24±4.49 | 51.73±4.34 | 56.00±3.53 | 50.26±1.73 | 61.38±1.72 | 42.19±1.16 | 46.29±1.29 |
| Meta-GPS | 65.19±2.35 | 69.25±2.52 | 61.23±3.11 | 64.22±2.66 | 52.16±2.01 | 62.55±1.95 | 42.96±2.02 | 46.86±2.10 |
| X-FNC | 69.32±3.10 | 71.26±4.19 | 49.63±4.45 | 53.00±3.93 | 52.36±2.75 | 63.19±2.22 | 41.92±2.72 | 46.10±2.16 |
| TEG | 72.14±1.06 | 76.20±1.39 | 61.03±1.13 | 65.56±1.03 | 57.35±1.14 | 62.07±1.72 | 47.41±0.63 | 51.11±0.73 |
| COSMIC | 73.03±1.78 | 77.24±1.52 | 65.79±1.36 | 70.06±1.93 | 52.98±2.19 | 65.42±1.69 | 43.19±2.72 | 47.59±2.19 |
| **STAR** | 77.77±0.10 | 81.24±0.98 | 68.60±0.63 | 73.53±0.49 | 61.15±0.62 | 68.39±0.95 | 50.89±1.03 | 57.97±0.88 |
| **STAR+MOEFL** | **84.00±0.90** | **82.96±3.00** | **71.09±0.57** | **74.82±1.04** | **66.93±0.19** | **70.39±0.80** | **53.38±2.02** | **57.94±0.57** |
| **GRACE** | 78.22±1.38 | 81.60±1.28 | 70.91±1.08 | 74.54±0.98 | 62.31±1.94 | 68.34±1.73 | 50.18±1.01 | 55.07±0.91 |
| **GRACE+MOEFL** | **85.41±1.66** | **85.89±1.48** | **79.84±1.22** | **81.58±0.84** | **67.92±1.80** | **70.10±1.80** | **54.45±1.02** | **57.94±0.57** |

In addition, we observe that the proposed method maintains comparable or lower variance in most settings, suggesting improved stability across different few-shot tasks.

## 4.2 Ablation study

### 4.2.1 Element Study

In this work, we propose a framework consisting of three key components: (1) motif-based virtual nodes for incorporating structural patterns, (2) cluster labeling for enriching motif representations with node feature information, and (3) TF–IDF-based edge weighting to regulate the influence of motif–node connections. In this section, we analyze the contribution of each component using GRACE as the backbone on the Cora dataset under the 2-way 5-shot setting.

The results are summarized in Table 5. Compared to the baseline GRACE model, introducing motif virtual nodes alone already leads to a modest improvement in performance, indicating that motif-level structural information provides useful inductive bias for few-shot learning. When cluster labeling is incorporated, we observe a slight additional gain, suggesting that integrating node feature information helps differentiate structurally similar motifs and improves their expressiveness.

Finally, combining all components, including TF–IDF-based edge weighting, achieves the best overall performance. This indicates that while motif connections introduce additional edges into the graph, properly weighting these connections is important to prevent noisy or overly dense interactions from degrading performance. Overall, the results demonstrate that the three components are complementary, and their combination leads to consistent improvements over the baseline.

### 4.2.2 Motif Node Study

In this section, we compare two strategies for enriching motif representations: increasing motif size (e.g., from 3-node to 4-node motifs) and applying cluster labeling to enhance motif expressiveness without enlarging the motif structure. All experiments are conducted using GRACE as the backbone on the Cora dataset,

Table 5: Ablation study on Cora (2-way 5-shot). Backbone: GRACE.

| Configuration | Motif VN | Cluster Label | TF-IDF | Accuracy (%) | Gain |
|---|---|---|---|---|---|
| GRACE (baseline) | ✗ | ✗ | ✗ | 86.19 ± 1.80 | − |
| Motif only | ✓ | ✗ | ✗ | 86.69 ± 1.74 | +0.50 |
| Motif + Cluster | ✓ | ✓ | ✗ | 86.63 ± 1.79 | +0.44 |
| **MoEFL (full)** | ✓ | ✓ | ✓ | **87.22** ± **1.70** | **+1.03** |

Table 6: Motif size and cluster labeling ablation on Cora (2-way 5-shot). Backbone: GRACE.

| Configuration | Node Size | Clusters | Motif Types | Accuracy (%) | Gain |
|---|---|---|---|---|---|
| 3-node | 3 | − | 1 | 86.89 ± 1.74 | +0.70 |
| 3-node + cluster | 3 | 4 | 20 | **87.22** ± **1.75** | **+1.03** |
| 4-node | 4 | − | 6 | 86.70 ± 1.69 | +0.51 |
| 4-node + cluster | 4 | 4 | 547 | 86.93 ± 1.70 | +0.74 |

while keeping the motif integration process and TF-IDF weighting consistent across all settings. The results are reported in Table 6.

From the results, simply increasing motif size provides limited benefits. Although larger motifs capture more complex structures, the number of distinct structural patterns remains relatively small (e.g., 1 for 3-node motifs and 6 for 4-node motifs without clustering). As a result, the performance improvement is marginal and even shows diminishing returns, suggesting that larger motifs may introduce redundant or noisy patterns that do not effectively contribute to model performance.

In contrast, cluster labeling consistently outperforms the structure-only approach. By incorporating node feature information, cluster labeling significantly increases motif diversity (e.g., from 1 to 20 motif types for 3-node motifs, and from 6 to 547 for 4-node motifs). This allows the model to distinguish structurally similar but semantically different patterns without excessively increasing structural complexity. As shown in Table 6, the combination of 3-node motifs with cluster labeling achieves the best performance, yielding a +1.03% improvement over the baseline. However, when applying cluster labeling to 4-node motifs, the number of motif types grows excessively, leading to over-fragmentation and the introduction of noisy patterns, which can negatively impact performance.

Overall, these results indicate that enhancing motif expressiveness through cluster labeling is more effective than increasing motif size, while also avoiding the computational and structural overhead associated with large motifs.

### 4.2.3 Cluster Method

In this section, we compare four different cluster methods, Kmeans (Lloyd, 1982), GMM (Weber et al., 2022), mini-batch Kmeans (Sculley, 2010) and agglomerative (Murtagh & Legendre, 2014). The backbone method is GRACE. The result is reported in Table 7.

From Table 7, we observe that our method is compatible with a variety of clustering approaches. All four clustering methods consistently improve performance over the baseline, indicating that the effectiveness of cluster labeling is robust to the choice of clustering algorithm. While the performance differences are relatively small, agglomerative clustering achieves the best result, with a 1.45% gain, suggesting that capturing hierarchical relationships between nodes may provide additional benefits. Overall, these results demonstrate that the improvement brought by MoEFL mainly stems from incorporating feature-aware clustering, rather than relying on a specific clustering technique. Further investigation into how different clustering methods affect performance across diverse datasets is left for future work.

### 4.2.4 Hyperparameter Sensitivity

In this section, we examine the sensitivity of our framework to key hyperparameters. Specifically, we consider two important hyperparameters that can affect the final performance. The first is the number of clusters,

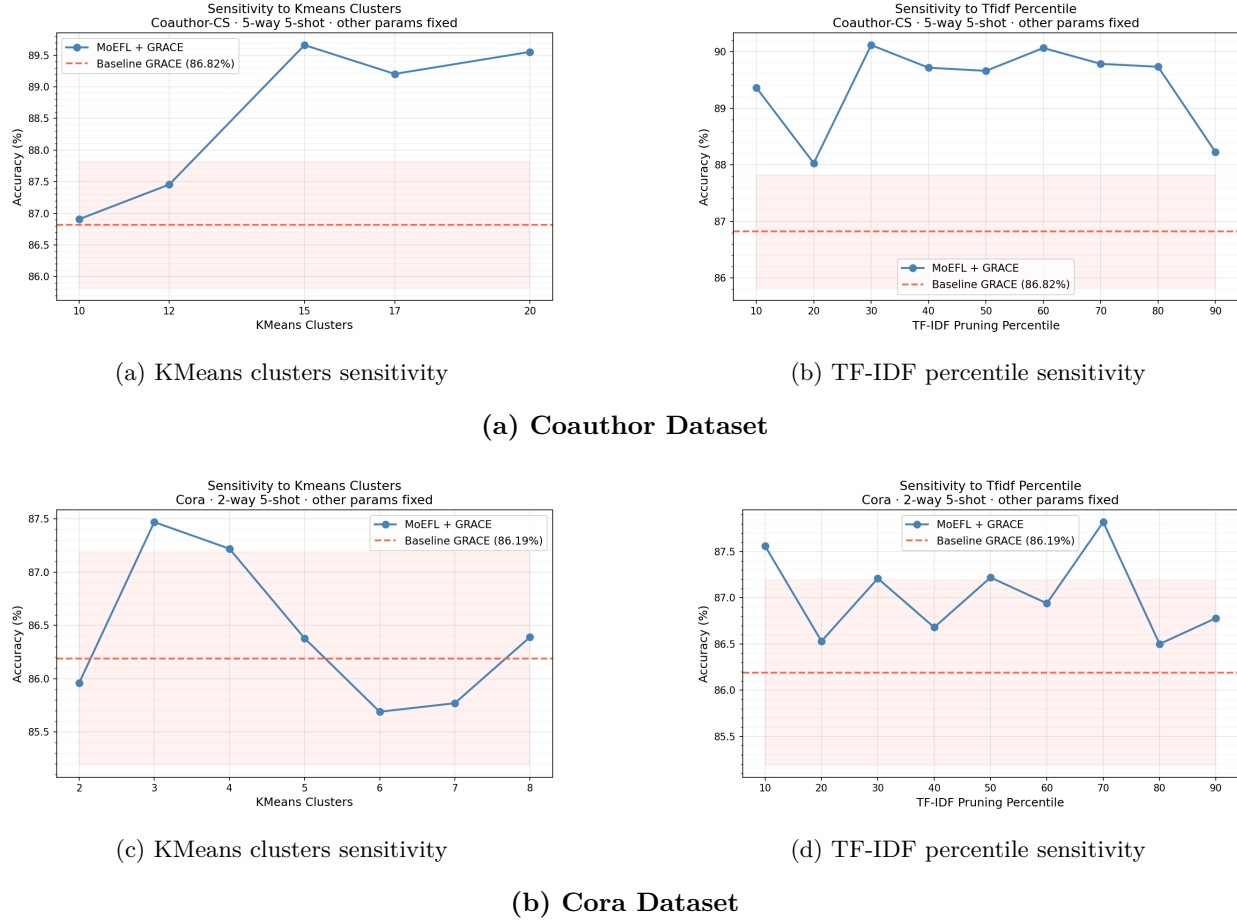

(a) KMeans clusters sensitivity

(b) TF-IDF percentile sensitivity

**(a) Coauthor Dataset**

(c) KMeans clusters sensitivity

(d) TF-IDF percentile sensitivity

**(b) Cora Dataset**

Figure 3: Sensitivity analysis across different datasets. Each row corresponds to a dataset, showing the effect of clustering methods and TF-IDF percentile thresholds.

Table 7: Comparison of clustering methods for cluster labeling on Cora (2-way 5-shot). Backbone: GRACE.

| Method | Motif Types | Accuracy (%) | Gain |
|---|---|---|---|
| GRACE (baseline) | – | $86.19 \pm 1.80$ | – |
| k-means | 20 | $87.22 \pm 1.70$ | +1.03 |
| GMM | 20 | $87.20 \pm 1.74$ | +1.01 |
| mini-batch k-means | 20 | $87.25 \pm 1.73$ | +1.06 |
| **agglomerative** | 20 | $\mathbf{87.64 \pm 1.73}$ | **+1.45** |

which directly influences the number of extracted motifs. The second is the TF-IDF pruning threshold, which determines how many motif–node edges are retained in the augmented graph. We conduct hyperparameter sweeps on two datasets, Cora (2-way 5-shot) and Coauthor-CS (5-way 5-shot), and report the results in Figure 3.

Overall, we observe that the proposed framework is generally not highly sensitive to either hyperparameter. For the number of clusters, the small graph will be affected more by the cluster number. optimal performance is achieved with a small number of clusters (3–4), while deviations from this range lead to slight performance drops. In contrast, larger graphs exhibit greater robustness to the choice of cluster number. On Coauthor-CS, configurations with 15, 17, and 20 clusters yield similar performance, indicating low sensitivity.

For the TF-IDF pruning threshold, the model demonstrates consistently strong performance across a wide range of values on both datasets. While moderate pruning levels generally achieve the best results, per-

Table 8: Motif extraction and injection time on different datasets. Motif extraction is performed only once as a preprocessing step, while motif injection is applied before each training and test phase.

| Dataset | Nodes | Edges | Extraction (one-time) | Injection |
|---------|-------|-------|----------------------|-----------|
| DBLP | 40,672 | 288,270 | 113.62 s | 4.08 s |
| ogbn-arxiv | 169,343 | 1,166,243 | 3592.00 s | 17.97 s |

formance degradation under extreme settings remains limited. This suggests that the proposed TF-IDF weighting mechanism is robust and does not require fine-grained hyperparameter tuning.

Overall, these results indicate that the framework is robust to hyperparameter choices. Both the number of clusters and the TF-IDF threshold can be selected from a broad range without significantly affecting performance, making the method practical and easy to apply in real-world scenarios.

### 4.3 Efficiency

In this section, we examine the impact of our framework on training and inference efficiency. We identify two main sources of computational overhead: (1) the motif extraction process and (2) the construction of the augmented graph. To evaluate efficiency, we conduct experiments on the two largest datasets used in this paper, DBLP and ogbn-arxiv.

The efficiency results in Table 8 demonstrate the scalability of our framework on the largest graphs in our experiments. In particular, on the ogbn-arxiv dataset with over 1.1M edges, motif extraction takes approximately 3600 seconds, while the injection process only requires a negligible 18 seconds. This highlights that the computational cost is dominated by motif extraction.

Importantly, motif extraction is a **one-time preprocessing** step. Once the motif structures are extracted, they can be reused across all training and evaluation tasks. Therefore, the additional overhead introduced during model training and inference is minimal, as only the motif injection step is required.

These results indicate that, although motif extraction can be time-consuming on large graphs, it does not impact the runtime of the learning process itself. As a result, our framework remains scalable and practical, even for large-scale datasets.

## 5 Conclusion

In this work, we propose a novel motif-embedding framework, MoEFL, to enhance the performance of graph few-shot learning models. We extract motifs from the original graph and represent each as a virtual node, with a learnable embedding. These virtual nodes are connected to the original graph nodes that belong to the corresponding motif. To enrich the extracted motif patterns, we introduce cluster labeling. Furthermore, we leverage TF-IDF scores to quantify the relevance between motifs and original nodes. Experimental results demonstrate that our proposed framework effectively improves the performance of graph few-shot learning models.

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
