# OpenReview forum: "Global Motif Embedding Meet Few-Shot Graph Learning"
_TMLR — Under review for TMLR_

### Review · Reviewer_hjxw · 2026-05-25

**Summary Of Contributions:**

This paper studies few-shot node classification and proposes MoEFL, a framework that injects motif-level information into graph few-shot learning models. The basic idea is to extract motifs from the original graph, treat motif types as learnable virtual nodes, and connect them with the original nodes that participate in the corresponding motifs. The authors also use feature-based cluster labels to make motifs more expressive, and introduce a TF-IDF-style weighting scheme for motif-node edges.
I find the idea interesting, since the paper tries to improve graph few-shot learning by using structural patterns in the graph, rather than only modifying the meta-learning or contrastive learning objective. The method is also fairly modular and is tested with STAR and GRACE on several datasets. That said, the current paper still leaves some important details unclear, especially how motifs are extracted and canonicalized, how the weighted motif-node edges are used, and whether the preprocessing uses information from test-class nodes. Also, some improvements are small or not consistent across all settings, so the claims should be made more carefully.

**Audience:**

Yes

**Audience Explanation:**

The topic should be of interest to part of the TMLR audience, especially researchers working on graph representation learning, few-shot learning, and structural inductive biases in GNNs. The idea of using motif information as an additional source of structural knowledge is natural but still not very common in graph few-shot learning.

**Broader Impact Concerns:**

I do not see any serious broader impact concerns for this work. The paper focuses on a general graph learning technique and tests it on commonly used benchmark datasets.
The only minor concern is more general to graph learning: if similar methods are later applied to social or user-related graphs, existing biases in the data may affect the model outputs. This does not seem to be a major issue for the current paper, but it may be worth a brief note from the authors.

**Claims And Evidence:**

Yes

**Claims Explanation:**

The paper provides a reasonable amount of experimental evidence to support its main claim that motif-level information can help graph few-shot learning. The authors evaluate MoEFL with two different backbones, STAR and GRACE, and report results on multiple datasets and few-shot settings. In many cases, adding MoEFL improves the performance of the backbone models, which suggests that the proposed motif augmentation is useful. That said, I think the evidence could still be made clearer and more convincing in a few places. Some of the reported gains are quite small or close to the standard deviation, and there are a few settings where the improvement is marginal or not present. Therefore, I would suggest that the authors avoid very strong wording such as “consistently improves” unless they provide significance tests or a more detailed discussion. And one ablation result needs a more careful explanation. In Table 5, the “Motif + Cluster” setting does not improve over “Motif only,” although the text suggests that cluster labeling brings an additional gain. This is not a major flaw, but the authors should revise the explanation so that it matches the table.

**Requested Changes:**

1.The paper would benefit from a more step-by-step description of how motifs are extracted and added to the graph.2.The TF-IDF weighting scheme is a useful idea, but the current description is somewhat brief. 3.Please state clearly whether motif extraction, node clustering, and TF-IDF statistics are computed on the full graph. 4.Make the claims slightly more cautious. 5.Add a few simple control experiments if possible.6.Provide more implementation details.6.Polish the writing.

---

### Review · Reviewer_de8x · 2026-06-30

**Summary Of Contributions:**

The paper introduces MoEFL, a modular, model-agnostic framework designed to improve few-shot node classification on graphs. It works by extracting structural motifs, expanding their diversity via feature-based cluster labeling, and injecting them into the graph as virtual nodes connected via TF-IDF-weighted edges. A dual-branch encoder with a learnable gate then fuses the original and motif-augmented representations.

***Key Strengths:***

- Provides a clean, plug-and-play mechanism to inject higher-order structural priors into few-shot graph learning.

- Thorough experimental validation across seven datasets, combining the framework with two strong baseline models (STAR and GRACE).

- Well-structured ablation studies that clearly isolate the benefits of motif nodes, cluster labeling, and TF-IDF weighting.

***Key Weaknesses:***

- Ambiguity surrounding the exact motif definitions, extraction pipelines, and scale of virtual nodes.

- Potential transductive data leakage during the clustering/motif extraction phase.

- Lack of evaluation in inductive settings and absent comparisons against simple linear-probing/inductive baselines.

**Additional Comments:**

Please increase the font size in Figure 1. It's very hard to recognize the content in it.

**Audience:**

Yes

**Audience Explanation:**

The TMLR audience includes many researchers focused on graph neural networks (GNNs), meta-learning, and representation learning. Addressing the scarcity of labels in graph tasks is a well-known challenge, and the proposed method of injecting higher-order structural priors via virtual nodes is a practical, lightweight solution. The plug-and-play nature of MoEFL makes it an attractive tool for practitioners building upon existing GNN architectures.

**Broader Impact Concerns:**

The unsupervised preprocessing step relies heavily on clustering and motif extraction, which appears to leverage global graph data. If applied to real-world networks containing sensitive or proprietary information (e.g., social networks, financial transactions, healthcare graphs), transductive access to the full graph topology could pose privacy risks. The authors should add a brief Broader Impact Statement discussing the privacy implications of global structure extraction and note the framework's potential limitations in dynamic or highly privacy-constrained environments.

**Claims And Evidence:**

Yes

**Claims Explanation:**

The authors provide substantial empirical evidence across multiple benchmarks and integrate their framework successfully with two established backbones. The ablation studies effectively validate the individual components of the MoEFL architecture. However, the evidence currently relies heavily on transductive settings. To make the claims fully convincing—particularly regarding generality and robustness—the authors need to address potential transductive leakage during the clustering phase and evaluate the framework in inductive settings.

**Requested Changes:**

- Explicitly state whether virtual nodes correspond to motif "types" or "instances." Provide the exact number of virtual nodes used per dataset and specify the chosen motif sizes.

- Explain the exact motif extraction/counting strategy (e.g., induced vs. non-induced, treatment of undirected edges) and how correctness and scalability are maintained on large datasets like ogbn-arxiv.

- Clarify if test-class nodes are used to build clusters and motifs during preprocessing. If so, provide an ablation quantifying the performance impact when clustering/extraction is strictly restricted to training classes.

- Evaluate MoEFL in an inductive few-shot node classification setting (removing cross-split edges) to support claims of framework generality.

- Report the training-time memory and runtime overhead introduced by the augmented graphs.

- Justify the choice of IDF=1/df. Compare this ad-hoc formulation against standard alternatives like log-scaled IDF, learned attention, or degree normalization.

- Analyze the learned motif embeddings. Report performance differences when embeddings are frozen at initialization versus randomly shuffled at test time.

---

### Review · Reviewer_N7aT · 2026-07-12

**Summary Of Contributions:**

In this work, the authors present MoEFL, a lightweight, model-agnostic module that incorporates structural information of motifs into graph few-shot learning pipelines. The extracted motifs are represented as learnable embeddings and attached to the graph in the form of virtual nodes which are connected to the original nodes that belong to the corresponding motif. After this, a "cluster labeling" phase is executed where the K-means (other cluster labeling algorithms can also be used) is performed on node features, which increases the number of distinguishable motif types while keeping the motifs small. The edge weights between each node and its motifs are set based on a TF-IDF style approach, such that the very common motifs are down-weighted. After this, the augmented graph goes through a second "augmented" encoder next to the original backbone encoder. The two resulting representations are fused using a learned gate. Motif embeddings are trained end-to-end during the meta-training process and are then frozen and used during the meta-test phase on novel classes. This method is tested as a plug-in on two recently proposed FSL backbones (GRACE, STAR) on seven datasets of different size.

# Strengths

In terms of the central intuition, the paper does seem to make sense. Considering motifs as a "vocabulary" through the lens of word embeddings and making use of cluster labels to augment the extremely limited space of structural motifs (only two patterns in the case of undirected 3-node motifs) seems to make sense. Conducting experiments on two distinct backbone graphs is a good practice for demonstrating the generality of the approach. In terms of empirical coverage, the paper has used seven distinct datasets ranging between 2.7K to 169K nodes across various configurations of N-way-K-shot. The ablation studies have also considered the three elements separately along with comparisons based on motif size vs. cluster labeling and different clustering algorithms.

# Weaknesses
"Consistent improvement" is not justified by the results. First, at least one cell (STAR on Coauthor-CS, 2-way-5-shot, Table 3) demonstrates MoEFL performing worse than baseline: 96.37 $\pm$ 1.97 vs 96.43 $\pm$ 0.47. Other improvements are within 1 SD of the baseline value; there's not a single sign of significance testing to justify the term "consistently". Also, many pieces of necessary information to reproduce the results are lacking: training setup, backbone/architecture/hyperparameters of backbone & augmented encoders, learning rate values, number of tasks/episodes, and not a single mention of any code being released. There's no attempt to isolate if the improvements are due to motifs specifically or simply due to extra capacity introduced by second encoder + extra virtual nodes. An important control experiment with random virtual nodes or structural non-motif features (e.g., degree) using same parameters is very much needed. The ablation and hyperparameter studies also run on a narrow slice of the full experimental grid, mostly Cora with GRACE, or Cora plus Coauthor-CS, so it's hard to know whether conclusions like "3-4 clusters is optimal" hold anywhere else.

**Audience:**

Yes

**Audience Explanation:**

In this case, few-shot learning and motif-based structure priors are both research areas of active pursuit, and a small, architecture agnostic module which could integrate with existing FSL frameworks would certainly prove useful if the experimental results held up. The use of the label propagation method to expand the small motif vocabulary (only two types of 3-node motifs) is fascinating in its own right, apart from the few-shot learning context.

**Claims And Evidence:**

No

**Claims Explanation:**

The primary quantitative statement, that "MoEFL consistently improves the performance of various graph few-shot learning methods" is contradicted by at least one finding reported (the STAR + MoEFL performing worse than STAR on Coauthor-CS 2-way-5-shot in Table 3). Most of the improvements reported are much smaller than the standard deviations in the experiments in which they were found. Nothing in the paper demonstrates the importance of the observed differences beyond noise. The ablations used to justify the architectural decisions are performed on a single dataset-backbone pair (Cora + GRACE), and this is not much given how wide the findings in the results table are. Also, because there is no baseline distinguishing between "extra capacity" and "motif-specific structure," it is hard to tell whether motifs matter at all.

**Requested Changes:**

1. State whether results are statistically significant in all comparisons of the main table, either by using a paired test on seeds/tasks or confidence intervals instead of standard deviations, and fix "consistently improves" when it doesn't.

2. Include a control ablation with the same addition of parameters or number of virtual nodes but without any motif semantics (using random virtual nodes or virtual nodes with simple properties such as degree).

3. Provide full reproducibility details about the encoder architecture and hyperparameters, training procedure, optimizer, learning rate, and number of episodes, as well as code if possible.

4. Conduct your ablation and comparison on motif size/clustering assignment on at least one more data set and backbone, aside from Cora and GRACE.

5. Make clear how the cluster labels and motif assignment types translate from the classes used for training to new, unseen classes at test time, as the K-means clustering depends on the feature distribution being clustered, and it's unclear if the test-time labels correspond to those of training.

6. Report total training-time overhead relative to the vanilla backbones (not just extraction and injection time) so the practical cost is clearer.